# The Potential Applications of Reinforced Bioplastics in Various Industries: A Review

**DOI:** 10.3390/polym15102399

**Published:** 2023-05-22

**Authors:** Uwei Kong, Nurul Fazita Mohammad Rawi, Guan Seng Tay

**Affiliations:** 1Bioresource Technology Division, School of Industrial Technology, Universiti Sains Malaysia, USM, Gelugor 11800, Penang, Malaysia; konguwei@student.usm.my (U.K.); fazita@usm.my (N.F.M.R.); 2Green Biopolymer, Coatings & Packaging Cluster, School of Industrial Technology, Universiti Sains Malaysia, USM, Gelugor 11800, Penang, Malaysia

**Keywords:** bioplastic, reinforced bioplastic, lignocellulosic, renewable resources

## Abstract

The introduction of bioplastics has been an evolution for plastic industry since conventional plastics have been claimed to cause several environmental issues. Apart from its biodegradability, one of the advantages can be identified of using bioplastic is that they are produced by renewal resources as the raw materials for synthesis. Nevertheless, bioplastics can be classified into two types, which are biodegradable and non-biodegradable, depending on the type of plastic that is produced. Although some of the bioplastics are non-biodegradable, the usage of biomass in synthesising the bioplastics helps in preserving non-renewable resources, which are petrochemical, in producing conventional plastics. However, the mechanical strength of bioplastic still has room for improvement as compared to conventional plastics, which is believed to limit its application. Ideally, bioplastics need to be reinforced for improving their performance and properties to serve their application. Before 21st century, synthetic reinforcement has been used to reinforce conventional plastic to achieve its desire properties to serve its application, such as glass fiber. Owing to several issues, the trend has been diversified to utilise natural resources as reinforcements. There are several industries that have started to use reinforced bioplastic, and this article focuses on the advantages of using reinforced bioplastic in various industries and its limitations. Therefore, this article aims to study the trend of reinforced bioplastic applications and the potential applications of reinforced bioplastics in various industries.

## 1. Introduction

Plastics are synthetic organic polymers mainly made by fossil fuel-based chemicals (petrochemicals), which can be used in various industries, such as packaging industries, healthcare industries (medical instruments), electronic industries (electronic goods), etc. There are around 8.3 billion tons of plastics that have been generated since the 1950s, and around 79% of them are not recycled, but left as waste in the environment [1]. Decomposition of plastics in the environment takes a few centuries, and, therefore, leaving plastic waste in the environment or landfilling the plastics are causing toxic pollution to the earth. Yearly, approximately 13 million tons of plastic waste have been thrown by humans into the ocean, which then harms marine lives [2].

Recycling is one of the approaches used to overcome the issues brought by conventional plastics, as the wastes could be recycled to be used as sustainable polymers [3]. The ultimate aim of recycling polymer is to develop eco-friendly recycled plastics, which meet performance and quality requirements [4]. Through recycling polymer, it can slow down the consumption of fossil-fuel based chemicals as the production of virgin plastics is slowed down. However, the recycled polymer products are lower in quality compared to virgin plastics due to the chemical impurity during recycling of the plastic wastes [5]. In addition, the process of reproducing the plastics involves the grading of polymers, washing, grounding, and extruding, which will lead to a certain degree of degradation, and this all limited the recycling cycle for plastics [6].

Bioplastics were developed as an approach to overcome the issue in the early 21st century. Rudin and Choi [7] defined bioplastics as commercial polymer products produced by renewable resources or natural sources. Under suitable conditions, some of the bioplastics are biodegradable, and some renewable resources made of bioplastics could be recycled through biological processes [8]. The common natural and renewable resources used for the synthesis of bioplastics are vegetable oil, starch, cellulose, protein, etc. Application of bioplastics has partly replaced the use of conventional plastics in various industrial applications, including packaging for food and others, medical instruments, hygiene, and agriculture. In recent developments, bioplastics can be applied in human bodies for medical usage, such as controlled drug delivery systems and therapeutic devices implantation [9]. The European Bioplastic has reported that the production capacity of bioplastics in the year 2022 is 1075 in 1000 tons, which is expected to increase to approximately 2453 in 1000 tons in the year 2024 [10]. As time goes by, awareness towards preserving and protecting the environment rises, and there are more demands for replacing conventional plastics with bioplastics. In the world market, around 1% of plastics produced annually are bioplastics, and the current trend shows that the bioplastics market has a continuous increment in various industries [11]. In this article, the aim is to study and identify the trend of reinforced bioplastic applications and the potential application of reinforced bioplastics in various industries through reviewing published journal articles focused on bioplastics and bioplastics reinforced by various types of materials.

## 2. Bioplastics

There are different classifications of bioplastics, which are mainly based on the biodegradability or the materials. Nevertheless, bioplastics could be classified into two major types in general, which are biodegradable bioplastics and non-biodegradable bioplastics. Biodegradable bioplastics refer to bio-resources-derived bioplastics, which are biodegradable, and these types of bioplastics include poly(lactic acid) (PLA), poly(hydroxyalkanoate) (PHA), and bio-based poly(butylene succinate) (Bio-PBS). On the other hand, non-biodegradable bioplastics include bio-based poly(ethylene) (Bio-PE), bio-based poly(propylene) (Bio-PP), and bio-based poly(ethylene terephthalate) (Bio-PET), which are non-biodegradable, although they are derived from renewable resources. However, in the study by Rosenboom et al. (2022), there are some petrochemical plastics, which can be biodegradable, such as poly(butylene succinate) (PBS) or poly(vinyl alcohol) (PVA), and they are also listed as bioplastics due to their biodegradability [12].

The common bioplastics used in the market are PLA, Bio-PE, Bio-PBS, PHA, poly(hydroxybutyrate) (PHB), etc. The three common ways used for synthesising bioplastics are chemical polymerization [PBS, poly(glycolic acid), etc.], use of natural polymers, such as starch and cellulose acetate, and bacterial polyester fermentation (PLA, PHB, PHA, etc.) [13]. PLA is a bio-based aliphatic polyester, which is a biodegradable and recyclable polymer, and it is a thermoplastic synthesis from a monomer named lactic acid that can be obtained from renewable resources, for example, sugarcane, starch, corn, food waste, and some other biomass [14,15]. PLA is well known for its good mechanical properties, compatibility, and biodegradability [16]. These advantages of PLA make it a suitable material as a replacement for conventional plastics in the applications of food packaging, construction, automotive, biomedical, and agriculture industries. The common methods to produce PLA include azeotropic dehydrative condensation, direct condensation polymerization, and lactide formation polymerization [15]. The direct condensation method required two steps, which is hydroxyl and carboxyl groups dehydration condensation to acquire low molecular weight (MW) PLA, followed by addition of esterification adjuvants and coupling agents in order to amplify the chain and modify PLA [17]. This synthesis method produces PLA with low impurities, which then needs flammable solvents to remove the byproducts within the product, which make it less preferable [18]. On the other hand, azeotropic dehydrative condensation used dibasic acids and glycols are used as solvents to replace adjuvants in the direct condensation method at high temperature with azeotropic distillation. This method could produce high MW of PLA, but the con of using this method is the remaining of solvents with catalysts [19]. Lactide formation polymerization, which also known as ring-opening polymerization (ROP), is one of the industrial production methods for high MW PLA, and it has mainly three types of mechanisms, which are coordination/insertion, cation use, and anion use [17]. In short, ROP starts with condensing lactic acid, followed by condensed water, mesolactic acid, and low MW polymer removal, recrystallization to obtain high MW pure lactide, and addition with those three mechanisms initiators to induce polymerization [17].

PBS is a biodegradable aliphatic polyester with excellent biodegradability, as well as thermal and chemical resistance, which made it suitable for various industry applications, such as food packaging, as well as textile industry usage as a filament and a split yarn [20]. The monomers can either be derived from petrochemical or bio-based chemicals (bacterial fermentation) [21] Basically, PBS is synthesised through a polycondensation reaction between 1,4-butanediol and succinic acid [22]. For bio-based monomers, succinic acid could be derived from glucose, which can be obtained from various sources of bio-feedstock through bacterial fermentation [23], while hydrogenation of succinic acid will produce 1,4-butandiol, which is required for PBS synthesis [24]. The synthesis of PBS requires two steps, which are esterification between diacid (succinic acid) and diol (1,4-butanediol), forming oligomers, followed by polycondensation at high temperature to form high MW PBS [21,22]. Table 1 summarises the common process of synthesising the aforementioned bioplastics.

In recent developments of bioplastics, there are few conventional plastics that have changed their raw materials to renewable materials, which transform them to bioplastics, and these plastics include Bio-PE, Bio-PP, and Bio-PET [11]. Compared to conventional PE and PP, Bio-PE and Bio-PP are synthesised using glucose, and glucose could be obtained from various feedstocks, such as sugar cane, starch crops, sugar beet, etc. At the same time, these bio-based plastics could be applied the same as conventional plastics, such as in packaging or film applications.

### 2.1. Reinforced Bioplastics

Recent research and development have achieved better mechanical properties of bioplastics, such as higher flexibility, strength, and thermal stability, but the problems and limitations still exist in achieving low cost bioplastic while having comparable mechanical properties with the conventional plastics in the current market [25]. In addition, shortcomings of bioplastics, such as weak mechanical properties, thermal stability, flexibility, and water permeability compared to conventional plastics, have limited their applications in various industries [26,27]. Hence, reinforcement of bioplastics with low cost and biodegradable natural material has become the focus of the current study in bioplastics. The common techniques used to reinforce bioplastics are blending another type of polymer into the bioplastic (polymer blend) and adding filler into the bioplastic. Polymer blending is a technique of adding another complementary polymer into bioplastics to overcome the shortcomings of bioplastics. For instance, Chotiprayon et al. (2020) blended thermoplastic starch (TPS) with PLA to reduce brittleness [28]. However, the study also stated that tensile and impact strength of the bioplastic blend are dropped. Hans-Josef Endres (2017) stated that the blend polymers are only thermodynamically miscible and stable when there is good compatibility between the blend polymers [29]. Nevertheless, the compatibility issue could be overcome with the addition of a compatibilizer. In other words, the polymer blend could overcome the limitations of bioplastics if the blend components have good compatibility.

Adding filler into bioplastics provides a variety of options for bioplastics, as there are many suitable materials, which are good in mechanical properties and strong in reinforcement. Generally, fillers have a good strength to weight ratio. However, they are unable to perform a task directly in an application in its current loose form, and they must be bonded or embedded in a matrix before they can be utilised in a specific application. On the other hand, polymer can be reinforced for strength and stiffness improvement for an application with an efficient stress transfer mechanism. Owing to this, the filler, either in particle or fibre form, is commonly used as reinforcement material for polymer matrix composites. For green development in bioplastics, natural resources and biomass are more suitable to be used as reinforcing materials. The most common green fillers used in bioplastics reinforcement are starch and fibre, which are two of the most abundant bioresources in the world. Starch can be obtained from plants, such as corn and potato, while fibre can be obtained from biomasses, such as bagasse and empty palm fruit brunch (EFB), and studies indicated that mechanical properties of bioplastics are significantly enhanced through the reinforcement of starch and fibre, which are abundant and cheap in nature [30,31,32]. Other than that, synthetic resources used in bioplastics reinforcement, including carbon fibre, glass fibre, and poly(propylene) (PP), give significant positive improvement to the properties of bioplastics [33,34]. Furthermore, chitosan and starch-based bioplastics reinforced with PP improve their mechanical strength at optimum conditions, as well as at suitable conditions, and the composite degraded more than 95% in 28 days, which shows the biodegradability of the reinforced bioplastics through using synthetic polymer as a reinforcement material [33]. In other words, synthetic materials also give significant increasing results in enhancing bioplastics, but the biodegradability of the reinforced bioplastics differs based on the compatibility and interaction between the materials with the bioplastics.

### 2.2. Reinforcing Materials

#### 2.2.1. Natural Resources and Biomass Materials

Natural resources and biomass are more suitable to be used as reinforcing materials for the sustainability and development of “green composites”. The most common natural resources used for bioplastic reinforcement are cellulose fibre, protein, chitosan, and clay. Cellulose is one of the abundant organic polymers from different sources, i.e., the agricultural sector, and it can be applied as polymer reinforcement in research studies, regardless of synthetic polymer or bioplastic [13]. Django (2016) stated that using cellulose fibre as a bioplastic reinforcement could reduce the carbon footprint while improving the biodegradability of plastic materials [35]. Studies show that cellulose reinforced bioplastics, which showed higher tensile strength, biodegradability, lower density, and lower permeability [36,37]. According to Satyanarayana et al. (2011), there were few types of cellulose to be used in reinforcing bioplastics, such as cellulose nanofibers and microcrystalline cellulose (MCC), and they required different treatments to process, which will determine the final properties of the cellulose-reinforced bioplastics [13].

In the study by Gao (2004), using nano-size fillers within plastic formulations helps in increasing interfacial interactions with the polymer matrix, which leads to better reinforcement [38]. Incorporating nanoparticle-sized clay in bioplastics increased the tensile strength of bioplastics while providing properties, such as homogeneous and transparent white colour [39]. On the other hand, chitosan is a hydrophobic and non-toxic polymer, which is rich in nature, and it can be found in the outer skeletons of shellfish, such as shrimp, crab, lobster, etc. Tan et al. (2022) stated that the nature of chitosan is suitable to be used as reinforcing material in overcoming the shortcomings of bioplastics, especially improving mechanical properties while increasing hydrophobicity to increase water resistance of bioplastics [40]. Lignocellulosic fibre is another popular ‘green’ option for reinforcing bioplastics, which has a high strength to weight ratio, a low cost, is lightweight, is sustainable, and is abundant in nature. It has been used for reinforcing bioplastics, as it has an effect in improving mechanical properties [26,41]. There is a large supply of lignocellulosic fibre in the market, such as kenaf, jute, sugar cane, and there is also biomass, which is rich in lignocellulosic fibre, namely, bagasse, empty fruit bunches from oil palm tree, and rice straw. Other than reinforcing bioplastics, using these materials could save cost for buying high-cost materials while preserving the environment through an increase in biodegradability and reuse of agricultural waste.

#### 2.2.2. Synthetic Materials

Synthetic resources used in bioplastics reinforcement include carbon fibre, glass fibre, and PP, which give significant positive improvement to the properties of bioplastics [33,34]. Glass fibre has high stiffness, but it has high strength, low density, relatively low cost, and high chemical resistance, making it suitable to be used for reinforcement purposes [42]. In the study of glass fibre reinforced bioplastics, it shows that the position of glass fibre during the loading process within the bioplastics is important, where placing it at the tension side of the bioplastic gives the best enhancement results concerning the strength properties [34]. Although reinforcing bioplastics with glass or carbon fibre produces higher strength products, it leads to worse environmental and health issues while processing. Reinforcing bioplastics using synthetic fibre indeed improves the properties of the bioplastics, but it also reduces biodegradability, which leads to another environmental issue [41,43]. PP is a synthetic polymer, which is slow in degradation in nature, but the degradation mechanisms could be accelerated through mixing with natural ingredients, for instance, chitosan and starch [33]. PP is well known for its mechanical strength, high thermal stability, and dimensional stability [44]. A study shows that chitosan and starch-based bioplastics reinforced with PP improve the mechanical strength at optimum conditions and at suitable condition, and the composite degraded more than 95% in 28 days, which shows the biodegradability of the reinforced bioplastics by using synthetic polymer as a reinforcement material [33]. In other words, synthetic materials also give significant increasing results in enhancing bioplastics, but the biodegradability of the reinforced bioplastics differs based on the compatibility and interaction between the materials and the bioplastics.

### 2.3. Comparison between Bioplastics and Conventional Plastics

#### Physical and Mechanical Properties

The differences between the vital properties commonly discussed in manufacturing ‘plastics’ products are compared. Table 2 outlined the comparison between reinforced bioplastics against conventional plastics and bioplastics. Degradability of plastic materials has been a concern in the industry for a few decades, as it takes an extremely long period for the degradation (10 to 1000 years), and the environment has to be suitable in order for the degradation mechanisms to occur [45]. The biodegradability of bioplastics depends on the materials used, and, for the biodegradable bioplastics, there are two types of materials, which are living organism derivatives (such as PHA) and biomass or bioresource derivatives (such as PLA and polysaccharides, including starch and cellulose) [13]. Nevertheless, for the biodegradable bioplastics, it also needs a suitable environment (presence of microorganisms, such as lower eukaryotic, bacteria, anaerobes, fungi, etc.) for the occurring of biodegradation mechanism [46]. Additionally, there are other factors affecting the biodegradation, such as chemical structures of bioplastics, UV exposure, temperature, present of different types of bacteria, and mechanical force influence, which will all affect biodegradation [47]. For reinforced bioplastics, the degradability also depends on the materials to produce the bioplastic and the reinforcement material, and thus, they also can be categorised into biodegradable and non-biodegradable reinforced bioplastics. However, similar to bioplastics, reinforced bioplastics also require suitable conditions for the degradation mechanisms to occur, and, therefore, waste management of the bioplastics is important, although the bioplastics could also degrade in nature, but this occurs over a longer period compared to those in a right environment for the degradation to take place. Research has shown that the addition of natural reinforcing materials (e.g., starch, cellulose, fibre, etc.) for bioplastics could have a faster biodegradation rate through increasing of hydrophobicity within the bioplastic matrix [48]. Non-biodegradable or synthetic materials (such as carbon fibre, clay, glass fibre, etc.) used for reinforcing would further reduce the biodegradability, although these materials give good effects in reinforcing bioplastics [41,43,49]. Still, the biodegradability of the reinforced bioplastics mostly depends on the compatibility and interaction between the materials with the bioplastics.

Sustainability of the plastic industry is vital for continuous development in the future for different types of application, and unsustainable conventional plastics, which consume petrochemical materials, will cause resource depletion in the future. Therefore, the industry urges us to replace conventional plastics with bioplastics that provide sustainability while reinforced bioplastics are developed to overcome the shortcomings of bioplastics. As can be seen from Table 2, bioplastics and reinforced bioplastics are both sustainable, but, for reinforced bioplastics, sustainability also depends on the reinforcing materials used because there are unsustainable materials, which can be used for reinforcement.

As shown in Table 2, other than the environmentally friendly property of bioplastics, the properties that determine the end applications of conventional plastics are much stronger than that of bioplastics. Although most bioplastics possess lower mechanical strength, there are some bioplastics which have high value in tensile strength, such as PLA. PLA has comparable tensile strength to conventional plastics, and it has been applied as an alternative to conventional plastics for years, but its brittleness has limited its application [50]. According to the standard of bioplastics and conventional plastics (Indonesia National Standard (SNI)), the acceptable tensile strength is in between 10–100 MPa and 24–302 MPa, respectively, and the tensile strength of cellulose acetate butyrate and most starch-based bioplastics did not meet the standard [51]. Factors affecting the tensile strength of polymers include molecular weight, crosslinking degree, and their crystallinity [52,53]. Therefore, tensile strength and mechanical properties of plastics are dependent on the chemical structure of the raw material and the polymer structure, either the chemical or physical structure. Petrochemicals have higher values of molecular weight, crosslinking degree (thermoset polymer), and crystallinity (for semicrystalline or crystalline polymer). As aforementioned, the addition of reinforcing material into bioplastics could increase the total crystallinity index if the reinforcing material consists of a crystalline structure. At the same time, it could promote more linkages while raising the molecular weight of bioplastics if both components (bioplastic and reinforcing material), which could interact and form a bridge, either through chemical or physical connection. Thus, reinforced bioplastics would have higher tensile strength and better mechanical properties than bioplastics, which is the result of the efficiency of the stress transfer mechanism, resulting from the formation of a bridge between the components. Besides, fibre-reinforced bioplastics could improve the properties of bioplastics through the uniform distribution of fibre throughout the bioplastic’s matrix. An even distribution of fibre within the bioplastics would show higher tensile strength after reinforcement, and properties of fibre, such as thickness and mechanical strength, could further improve the tensile strength of bioplastics [31]. Jethoo (2019) studied thicker fibre-reinforced bioplastics, which showed worse results due to difficulty in mixing and distribution of fibre in the matrix [31]. This is believed to be a result of the aspect ratio of the fibre, which does not mean that the minimum range to provide the reinforcing effect to the bioplastic is absent.

In a study using lignocellulosic fibre as reinforcement in soy-based bioplastics, the stress and strain increased 160% (from 1.8 MPa to 30 MPa) and 115% (from 0.8 MPa to 2.0 MPa), respectively, while the Young’s modulus (GPa) showed an increment of 50% at 5% loading [54]. In an article study on different natural fibres to reinforce PLA, namely, sisal, elephant grass, hemp, jute, ramie, and kenaf fibre, the tensile strength of the reinforced bioplastics showed various ranges of increment, and the highest increment shown is up to 81.8%, where the bioplastic is reinforced by kenaf fibre at 70% loading [55]. The tensile and flexural strength of 70%-loaded kenaf fibre-reinforced PLA increased from about 40 MPa to 223 MPa, and it also displayed a change from 70 MPa to 254 MPa [56]. However, in some cases, beyond a certain loading percentage of the reinforcing materials, these changes would cause negative effects on the biocomposite. For instance, more than 40% EFB loading in PBS will have significant deduction on mechanical strength because of the insufficient fibre wetting, leading to an interfacial adhesion issue between the polymer matrix with the fibre [57]. Kumar et al. (2021) also listed the decrement of natural fibre-reinforced PLA after certain loading, and the tensile strength started to decrease, and there was a drop under the original value of bioplastics: a PLA/ramie and PLA/flax drop after 30% loading, a PLA/hemp drop after 40% loading, and a PLA/sisal and a PLA/elephant grass drop after 20% loading. By contrast, PLA/kenaf did not showed any decrement, even at 70% fibre loading [55]. In the same study on different natural fibre-reinforced PLA, the flexural strength of the reinforced PLA showed the same increment behaviour, where the flexural strength of reinforced PLA increased to a certain value and dropped. Different reinforcing materials and bioplastics will cause variation in maximum reinforcing materials that can be added into the formulation.

As stated in Table 2, brittleness and water permeability of bioplastics is high, while it has lower thermal stability compared to conventional plastics. Brittleness of bioplastics has been one of the largest limitations, for instance, despite PLA having high tensile strength and thermal stability. The brittleness property has limited applications in various industries. To resolve this, plasticiser is often added to improve the brittleness, and it indeed enhances the flexibility, but it also brought other negative effect to the bioplastics, for instance, mechanical strength and water absorption [57,58,59,60]. Harmaen et al. (2013) added triacetin on EFB-reinforced PLA, and the result showed increment in tensile modulus (GPa) from around 1.7 GPa to 2.6 GPa, but the tensile strength showed a decrement of around 43.5% [61]. In order to overcome the brittleness issue with the negative effect of plasticiser, reinforcing materials (fillers) are used. A study, adding castor oil as plasticiser in bagasse-reinforced starch-based bioplastics, enhanced the elongation and tensile strength of the bioplastics by around 311.8% and 42.8% respectively [32]. EFB-reinforced PLA showed decreased stiffness from 38.7 MPa to 33.4 MPa (deduction around 13.7%) at 50% EFB loading [61]. Reinforced bioplastics have lower brittleness, which could widen their applications [62]. In addition, the poor permeability and thermal stability of bioplastics could be improved to a comparable level with conventional plastics after reinforcement [62,63]. Gamero et al. (2019) reported a high processing temperature (120 °C) of 5%-loaded lignocellulosic fibre-reinforced soy-based bioplastics, which could have higher stress and strain because the reinforced bioplastics did not degrade at high heat, which indicates that the increment in thermal stability of the reinforced bioplastics [54]. Furthermore, the 5% lignocellulosic fibre-reinforced soy-based bioplastic shows deduction of water uptake capacity at approximately 185% [54].

## 3. Applications of Bioplastics and Reinforced Bioplastics in Various Industries

### 3.1. Healthcare Industry

In the healthcare industry, different types of thermoplastics have been employed due to their excellent properties to serve in applications. The high usage of plastics includes plastic medical instruments, such as IV tubes, surgical gloves, blood bags, syringes, and packaging of medical instruments, which could ensure hygiene conditions. Nowadays, biodegradable bioplastics have been applied in healthcare applications, such as controlled drug delivery systems and therapeutic device implantation [9]. For any medical instrument, it needs to undergo a sterilization process (e.g., high temperature steam sterilization, ethylene oxide (EtO), gamma irradiation), and it has a high chance to contact with various chemicals or body fluids, and these conditions often lead to an increment of biodegradation rate and a reduction of molecular weight of polymers [64]. In other words, the materials used in such applications should have high resistance against different types of chemicals and sterilization processes while maintaining the functionality and safety of the instruments [65]. Biodegradable bioplastics, such as poly(lactic-co-glycolic acid) (PLGA), PLA, and Poly(ε-caprolactone) PCL, are used in medical applications, including tissue engineering, as it is biodegradable by fungi and bacteria within humans’ bodies [66]. For the biomedical applications, bioplastics are used to be applied in human’s bodies. This often requires that the bioplastics possess biodegradability and non-toxicity, so the bioplastics application did not require an additional process to be removed from the body, and it did not bring harmful effect to applicants.

Bano et al. (2018) stated that biodegradable bioplastics, i.e., PCL, PLA, and PLGA, have various applications in the industry because of their bioresorbable and biocompatibility properties, but the mechanical properties of these bioplastics have limited their applications, such as bone regeneration or replacement, which the bioplastics could not support, and some tissue engineering applications, due to low mechanical properties, could not match the tissue properties [66]. In addition, it has been reported that most of the biopolymers have lower resistance against high temperature and the commonly used chemicals in the industry [65]. Metal medical instruments could not be replaced with bioplastics because of the shortcomings of bioplastics, which have limited the applications of bioplastics in the healthcare industry. Therefore, research on blending bioplastics with other materials to reinforce the properties is essential to ensure the properties of the reinforced bioplastic, which could be employed in specific applications and fulfil the requirements. In Advanced Drug Delivery Reviews (2016), applications of using reinforced PLA include gene delivery, tissue engineering, implants, shape memory, and controlled-release drug delivery [67]. Liu et al. (2011) reinforced PLA through blending with ethylene vinyl acetate copolymer (EVA) to produce paclitaxel-eluting stent coatings, which could modulate the drug release amount and rate through adjusting the PLA amount in the formulation [68]. Saini et al. (2016) stated that the use of PLA as one of the components in tissue engineering scaffolds gives better processing property, and using reinforced PLA, i.e., poly(glycerol sebacate) (PGS)/PLA blend, incorporated the properties of faster degradation and better wettability, which could further improve the biodegradability and compatibility with the tissue recovery period [67]. Besides, reinforced bioplastics are widely applied in packaging for healthcare products, including medical and personal care. As bioplastics are proven to be hygienic for healthcare product packaging, reinforced bioplastics further improve the functionality of the packaging. Reinforced bioplastic packaging, with improved mechanical properties, are higher in resistance against mechanical damage that led to tearing or breaking of the packaging, where these properties could further prevent virus and bacterial contamination on the medical instrument packed with the reinforced bioplastics. Chitosan, chitin, and polysaccharides impart anti-bacterial properties to the packaging while increasing the shelf-life of packaging [69,70].

Research by Dan Kai et al. (2018) suggested that lignin-reinforced bioplastics, which have high antioxidant activity, could be used as antioxidants to protect humans from oxidative stress, protect skin from radiation and pollutants, and promote regeneration of cartilage tissue [71]. Additionally, reinforcement, using material, such as lignocellulosic fibre and lignin, further imparts antimicrobial function to the bioplastics [41]. In the future, regarding reinforced bioplastic application in the healthcare industry, it is possible for bioplastics to be applied in biophotonic applications, where biophotonics is the technology of utilising or producing photons and light to image, identify, and analyse cells and tissues through the light properties of living structures, and this application could ease tissue engineering by using reinforced bioplastics in human [9]. Furthermore, a study shows that clay-reinforced PLA-based bioplastics have great potential in packaging for personal care and medical products, as the migration from the reinforced bioplastic is within the standard, and the mechanical properties could resist the working environment of these products [70,72].

### 3.2. Electrical and Electronic Industry

Nowadays, bioplastic has a wide range of applications in electrical and electronic industry (E&E industry), including bioplastics conductors, which are presently known as solid polymeric electrolytes (SPEs), which are used for the development of electrochromic devices, batteries, diodes, fuel cells, etc. [73,74]. Other than that, bioplastics have high usage in assemble part for demanding consumer products, such as casing for computer elements and mobile phones, speakers, mice for computer, and vacuum cleaners [8]. Bioplastics can be applied as membranes for electroacoustic devices, reinforcement for electronic paper, and for water treatment [75]. In the field of electroacoustic devices, bioplastics possess similar vibration intensity as aluminium and titanium diaphragms, with the ability to make deep bass notes and clear trebles tones [76]. Bozó et al. (2021) stated that thermal and electrical properties of conductive bioplastics had limited their application in replacement of critical metals in the industry [77]. Thermal and electrical properties are indeed a vital issue in applying bioplastics in the industry. Other than being applied as assembly parts, bioplastics have limited applications in the industry, which is due to the main requirement in the industry being electrical and thermal conductivity. Commonly, without specific functionality incorporation or any reinforcement, most conventional plastics are electrical and thermal insulator, while bioplastics have low electrical and thermal conductivity, as well as low thermal stability, which further limited their application in the E&E industry [78]. Reinforcing bioplastics with suitable materials could incorporate and strengthen the electrical and thermal conductivity properties, which make it have wider applications.

Reinforced bioplastic imparts additional effects, including conductive, magnetic, optical and electrical properties, and thermal stability enables it to have a wider application in the industry [79]. Reinforcing bioplastics using carbon nanotubes (CNT) and cellulose nanofiber showed higher mechanical, electrical, and thermal properties as compared to a non-reinforced bioplastic, which is more suitable for electronic applications [79,80]. Cellulose nanofiber-reinforced bioplastics and CNT-reinforced bioplastics have various applications in the industry, such as flexible photovoltaic cells (solar cell), sensors, advanced electronics, and as substrates used in roll-to-roll fabrication techniques [79]. In previous roll-to-roll techniques, bioplastics as the substrate faced difficulty, as they possess a high coefficient of thermal expansion (CTE), where the dimension of the bioplastic changed upon working under relatively higher temperature, but, with the addition of cellulose nanofiber into the bioplastics, it improved the performance of the CTE, which allowed the applications of reinforced bioplastics in the roll-to-roll technique [79]. Besides, there are three-dimensional printing filaments in the market using graphene-reinforced PLA instead of conventional plastics, which give the advantages of quicker cooling rate due to high thermal conductivity, lower deformation, and biodegradability [81].

Graphene has excellent thermal, mechanical, and electrical properties, and it is incorporated in bioplastics, which greatly enhances mechanical properties of bioplastics while maintaining high flexibility and imparting electrical conductivity [82]. The graphene-reinforced PLA was also reported to have the potential to be used in orthopaedic and scaffold applications [81]. Besides, carbon fibre, which has high physical, mechanical, and thermal properties, are incorporated in PLA to impart excellent electrical conductivity and electromagnetic interference in the bioplastics, which then made it have the ability for electromagnetic interference (EMI) shielding [83]. A study conducted by Bozó et al. (2021) stated that carbon-based conductive materials have high electrical conductivity and could be used as reinforcements for bioplastics, and they might be suitable to be an alternative for the metals in applications that do not require extremely high electrical conductivity [77]. Electronic devices, including sensors, communication devices, transistors, inductors, electromagnetic shields, and capacitors, are also listed as potential reinforced bioplastics in the future of E&E industry [13].

### 3.3. Architecture and Construction Industry

Usage of plastics for architecture and construction purposes was started a few decades ago. The common plastic usages in these industries are pipes, insulation, floor coverings, cables, etc. Wall cladding, geotextiles, façade elements, and pipes are examples of bioplastic material applications in the architecture and construction industries. Conventional plastic applications in the industry, including textile fleece, cables, or floor coverings, require high strength to resist against different heavy workload conditions. Bioplastic materials have great application potential in the industry, but the cost is high due to the fact that better quality bioplastics mostly require extra cost in processing, and the performance of conventional bioplastic in the industry is not suitable and is insufficient to apply in the industry [84,85]. Ivanov and Stabnikov (2017) stated that the application of biodegradable bioplastics can give benefits to the industry, such as environmental and bio-economic sustainability, reduction of construction waste disposal cost, and temporary construction excavation costs [86]. Friedrich (2022) stated that the selection of construction materials is preferable for conventional plastics, as the properties of bioplastics, for instance, mechanical strength, resistance against biodeterioration agents, and life-span, are not guaranteed [87]. Besides, textile fleece in the industry usually prefers membrane materials, and, in this application, conventional plastics, which have higher strength and flexibility, are more preferable than bioplastics [88].

Mechanical strength of bioplastics could be enhanced through the addition of reinforcing materials, which enable application of reinforced bioplastics in the industry. Bioplastics reinforced with different structural materials provide benefits, such as being water resistant, being lightweight, having high stability in load resistant, having a low cost, having durability, and having ease of processing as construction materials in the industry [89,90]. Reinforced bioplastics are commercially available to be used as stabilisers for earthen construction materials, which previously used cement, and reinforced bioplastics can improve strength, durability, recyclability, and water resistance [90]. There was a study on using gel-type bioplastics reinforced with natural fibres and xanthan gum, which shows higher mechanical properties and lower shrinkage value [90]. Moreover, lignocellulosic fibre-reinforced bioplastics are commonly used in construction products, for instance, window frames and doors, because of their hydrophobic property and resistance to biodeterioration [89]. Lignocellulosic fibre-reinforced bioplastics have the advantages of strong mechanical properties, being elastic, and being biodegradable. Other than that, bioplastics, such as PLA, are reinforced with clay or CNT, and they showed improvement in mechanical properties, such as tensile strength, scratch resistance, and break elongation [86].

One of the potential applications of reinforced bioplastics is the high strength characteristics to be applied in construction formwork [85]. Besides, reinforced bioplastics are potential materials for construction textiles to replace conventional plastics, as fibre=reinforced bioplastics possess the quality and properties required, as stated by Friedrich (2022) [87]. Other than that, reinforced bioplastics can also be used as insulation for partitions and walls in temporary constructions, or they can be used for building temporary constructions [91].

### 3.4. Agricultural Industry

In the year 2017, the total plastics used in the agricultural industry were recorded as 6.96 million tons [92]. Cultivation films, greenhouses, tunnels, pesticide containers, storage bags, mulching, and pots are examples of plastic applications in the industry. There are wide applications of plastics in the industry due to their durability, water resistance, lightweight nature, and protective properties, and these properties can maximise crop yields [93]. Bioplastics also have applications in industry, such as in pots, seedling trays, mulching, and polymer-coated fertiliser [93,94,95]. Seedling trays tend to disintegrate in the soil after the sprouting and the growth of seeds, and the biodegradation of seedling trays would not emit harmful chemical to the soil or be absorbed by the plants [96]. One thing to highlight in these applications is the bioplastic mulching films, which are also known as biodegradable plastic mulches (BDMs) in the industry. Materials used for producing BDMs include cellulose, starch, PLA, and PHA, and these BDMs are ploughed into the soil after being used, as they can be degraded by microorganisms in the soil, and decomposition of BDMs into the soil can improve soil quality, led by enhancing microbial activity [97]. Other than that, PHA also have been applied as carriers for insecticides, crop protection films, fertilisers, and seed encapsulation [98].

A study on bioplastic-reinforced with EFB in producing high quality BDMs shows that fibre-reinforced mulch film could strengthen the film while helping the plants in adapting climate changes [57,99]. In addition, lignocellulosic fibre-reinforced bioplastics accelerate the polymeric matrix degradation rate in soil, which made reinforced bioplastics more suitable to be used as plant nursery bags or pots in the agriculture industry [100,101]. Besides, reinforced bioplastics also have been used as packaging for agricultural products, as the porous property helps in better air flow, which keeps the freshness, and the enhanced flexibility and strength of bioplastics improve the durability of the agricultural products packaging [102]. Using natural resources in reinforcing bioplastics for plant nursery in the industry would provide a similar function to bioplastics in BDMs, which increase the soil quality when the reinforced bioplastics decomposed in the soil. However, the elements and gases emitted have to be fully studied to avoid any disadvantages to agricultural crops.

### 3.5. Packaging Industry

Data provided by the European Association of Plastics Recycling and Recovery Organizations (EPRO) in 2020 show that usage of plastics in the six largest European countries of the packaging industry is around 40.5%. In the year 2021, use of bioplastics as packaging material is about 48%, which is approximately 1.15 million tons of global bioplastic production [1]. The common bioplastic applications in the packaging industry are bags, films, and wraps. In the beverage packaging market, non-biodegradable bioplastics are used, for instance, Coca-Cola designed PlantBottle^®^ synthesis from bio-based ethylene glycol (made up of 30% Bio-PET), and PepsiCo used switchgrass, corn husks, and pine bark to produce 100% Bio-PET for their product packaging [103,104]. Other than that, bioplastics, such as PHA, starch, and PLA, are also often used in food packaging, but these types of bioplastics have lower thermal stability, which leads to difficulty in forming the packaging product [104]. Bioplastics are low in mechanical and permeability properties for the use of packaging, and reinforcing bioplastics could overcome these shortcomings [105].

Reinforced bioplastics using agricultural waste materials, such as EFB and rice straw, are used for fresh fruit packaging, and the result shows that the reinforced bioplastics could lengthen the storage life of fresh fruit through improving the permeability properties of bioplastics, which promote better air flow for fresh fruit transpiration [102]. One of the reinforced bioplastics applications in the industry is the packaging of food. For the food packaging industry, bio-based materials are preferable in reinforcing bioplastics due to safety and environmentally friendly issues. Different from packaging for fresh products, food packaging requires low permeability to prevent food rotting from moisture or gases penetrating [106]. The applications of reinforced bioplastics in food packaging include bottles for dairy products, containers, films, dishes, and bags for takeaway [106]. Other than that, fibre-reinforced PHBV was also reported to be used for active food packaging, which can increase the storage time and impart antimicrobial property to the packaging film, resulting in a higher possibility of preventing the rotting of food [107]. In addition, cellulose-based films reinforced with clay also show antimicrobial properties in packaging film developments with better properties in gas permeability and thermal stability [70]. Moreover, poly(butylene adipate-co-terephthalate) (PBAT) reinforced with thermoplastic starch (TPS) bioplastic films could reduce mold and yeast count while preventing the darkening of food, which further proved that reinforced bioplastics possess antimicrobial properties [104].

Gadhave et al. (2018) reported that the addition of acetylated starch in corn starch-based films has higher thermal stability with reinforced resistance against sealing, making it a suitable material for heat sealing packaging [108]. In the Handbook of Bioplastics and Biocomposites Engineering Applications (2011), it was stated that reinforced bioplastics could be the material for optoelectronic packaging in the future [13]. Optoelectronic packaging (OEP) is a type of electrical packaging that involves providing mechanical support, as well as electrical and optical connection, to electronic devices for the device’s continuous functionality, and the most commonly used material for this application is epoxy [13,109]. The environmentally friendly property of reinforced bioplastic also make it a suitable material for airline cosmetics packaging [110].

The summary of the applications of bioplastics and reinforced bioplastics is shown in Table 3. Table 3 depicts the application of different types of reinforced bioplastics in the replated industries. It can be seen that reinforced bioplastics have received considerable attention in relevant industries, which may be attributed to their acceptable characteristics in the application. The potential applications of reinforced bioplastics could be seen through reviewing of the applications of bioplastics and reinforced bioplastics in various industries.

## 4. Advantages and Limitations of Reinforced Bioplastics

One of the aims of development of bioplastics is to achieve the sustainability goal in the plastic industry. However, due to the weakness of bioplastics as aforementioned, the applications of bioplastics are limited in various industries. In the recent definition of bioplastics, biodegradable fossil-fuel based plastics also have been listed as bioplastics, and there is also non-biodegradable bioplastic. Thus, the biodegradability of reinforced bioplastics did not differ much from bioplastics, as it depends on raw materials. Some reinforcing materials could improve the biodegradability of the bioplastics, but using non-biodegradable or synthetic materials in reinforcing bioplastics could also reduce the biodegradability [48]. Sustainability of reinforced bioplastics depends on the selection of raw materials that could attain the application of the needs and preserve the environment. Nonetheless, as the industry nowadays tends to develop ‘green’ technologies, using sustainable and biodegradable materials are preferable in terms of reducing carbon footprints, preserving non-renewable resources (fossil-fuel resources), having less toxicity, and reducing plastic wastes, which lead to pollution that is harmful to the environment and human beings [75]. Bioplastics (i.e., PLA, bio-PE, bio-PP, bio-PBS) and reinforcing materials (i.e., cellulosic fibre, EFB, cellulose, starch) could be obtained from various feedstocks and biomass, which further reduced and utilised the wastage of various functional materials.

There are various types of reinforcements on bioplastics to achieve comparable properties with conventional plastics in order to reduce the applications of conventional plastics in various industries. The main advantage of reinforced bioplastics compared to bioplastics is properties enhancement. As aforementioned, through reinforcing bioplastics with different materials, such as lignocellulosic fibre, synthetic fibre, cellulose, starch, clay, CNT, or blending with other polymers, the physical, chemical, thermal, and mechanical properties of the bioplastics could be improved to be able to substitute the usage of conventional plastics. There are a lot of reinforced bioplastics applications in various industries due to better performance—for instance, active food packaging, advanced electronics, human tissue engineering, or replacement for lower workload construction materials, which previously could not be performed by bioplastics. In addition, the reinforcing agents could impart additional functional properties to bioplastics, including electrical conductivity, antimicrobial, antibacterial, and antioxidant properties. These additional functional properties widen the applications of plastic material to be used in biotechnology, conductive materials, or as replacement for metal and construction materials in the future.

Using suitable reinforcing agents could improve the biodegradability of bioplastics, but this is only limited to biodegradable reinforced bioplastics. The application of reinforced bioplastics could minimise the damage from littering of ‘plastics’ to the environment. It might cause more ‘plastics’ littering problems, as when the consumers know the biodegradability of the reinforced bioplastics-made products, wherein they might dump more waste, regardless of the type of plastics. In addition, although the reinforced bioplastics are biodegradable, this biodegradation still takes a certain amount of time. During the biodegradation process, which depends on the biodegradation environmental parameters, it might accidentally be consumed by marine fauna before it is degraded. As for biodegradable reinforced bioplastics, the biodegradation mechanisms often occurred under suitable conditions, and if the reinforced bioplastic wastes were just left in nature without suitable conditions, the biodegradation rate would be slow, while non-biodegradable reinforced bioplastics would give the same environmental pollution to the planet as conventional plastics if there is no proper plastic waste management [111]. Mtibe et al. (2021) also stated that some of the properties of bioplastics and its composite (reinforced bioplastic) would degrade after a certain time of recycling due to degradation of the polymer chains. During reprocessing the wastes into new products, there might be contamination of conventional plastics that could further degrade the polymers due to the incompatibility, and to avoid this situation, extra cost is included to separate different types of plastics [111]. In layman’s terms, management and recycling of the reinforced bioplastics waste has become one of the limitations to continuously apply reinforced bioplastics in the industries, as poor management will still cause a similar pollution issue to conventional plastic.

In the history of plastics development, reinforced bioplastic is a relatively new technology, and there are only limited studies on it. In a review study by Boey et al. (2022), the parameters (reinforcing agent loadings) in reinforcing bioplastics should be in optimum conditions to obtain the desired matrix and properties [112]. For production of reinforced bioplastics, there are various types of bioplastic materials and reinforcing agents, where the reinforcing agents are not necessarily compatible with the bioplastics, and, at a molecular level, most biopolymer blends are incompatible [111,113]. Compatibility is an important factor that affects the end-performance and mechanical properties, as it affects the interfacial adhesion within the polymer matrix. Blending of incompatible materials will results in poor interfacial adhesion that leads to worse properties and performance at end-use. Either adding compatibilizer into the formulation or using compatible materials in reinforced bioplastics formulation are solutions that could be used to overcome the compatibility issue of reinforced bioplastics. However, adding extra components might affect the matrix properties later, which causes another issue in synthesising new types of plastic products. These issues are highly related to the lack of studies in reinforced bioplastics. The combination between different materials in synthesising reinforced bioplastics is countless, and thus, there should be more study on interfacial properties of bioplastics to understand the reaction between different reinforcing materials with bioplastics. Continuous study and development of reinforced bioplastics are necessary to explore more potentials of applications of reinforced bioplastics in various industries.

Figure 1 illustrates the variety of bioplastics and fillers with the applications of the reinforced bioplastics in various industries. It summarises the applications of reinforced bioplastics, and its advantages and limitations are discussed in this study.

## 5. Conclusions

Bioplastics have been developed as green alternatives for conventional plastics, which can be classified into two types, which are biodegradable and non-biodegradable, depending on their raw material and the type of plastic that is produced. Bioplastics could be derived from various feedstocks or biomass, i.e., food waste, sugarcane, plant oil, starch crops, and sugar beets. Nevertheless, their poor mechanical strength, water permeability, thermal stability, and brittleness have limited their applications. Reinforcing agents, such as lignocellulosic fibre, starch, and cellulose, are added into bioplastics to improve their properties. At optimum loading of reinforcing materials, the mechanical and thermal properties, including water permeability and stiffness of the bioplastics, could be enhanced to meet industry performance and quality requirements. Through the reinforcement, the end-performance and properties of the bioplastics are on-par with conventional plastics, which later could be applied in various industries. Additionally, with the optimum conditions in reinforcement, reinforced bioplastics could surpass the application of conventional plastics that widen their potential applications, including conductive materials or as replacements for metal and construction materials in the future, as well as functional materials in biotechnology field. In short, the advantages of reinforced bioplastics are the sustainability and potential applications in the future through properties reinforcement. However, the management of reinforced bioplastics has become a limitation, as the biodegradability and recyclability of reinforced bioplastics differ and depend on the environmental conditions. Additionally, there are limited studies on the combination and compatibility between reinforced bioplastics materials, as there are a lot of variances of materials and different properties. Thus, further research and study on reinforced bioplastics is necessary to widen the applications of reinforced bioplastics. Future studies on reinforced bioplastics could focus on compatibility of bioplastics with different variance of natural reinforcing material, potential of various bioresources as reinforcing materials, and the properties of reinforced bioplastics.

## Figures and Tables

**Figure 1 polymers-15-02399-f001:**
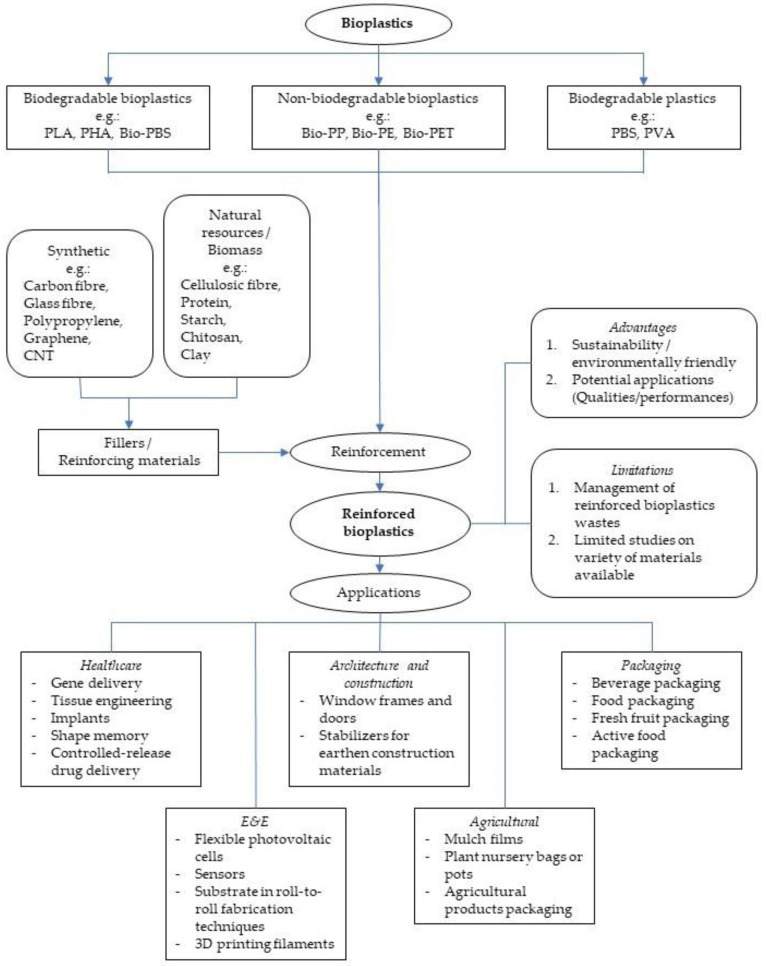
Summary of potential of reinforced bioplastics.

**Table 1 polymers-15-02399-t001:** Common synthesis process of bioplastics (reference from: Siracusa et al. [11,17,21]).

	Synthesis Process	
	Process	Product
Bio-PE	Anaerobic fermentation of glucose	Bioethanol
	Distillation of bioethanol	Azeotropic mixture of hydrous ethanol and vinasse
	Dehydration at high temperature with catalyst	Bio-ethylene
	Polymerization of bio-ethylene	Bio-PE
Bio-PP	Fermentation of glucose	Iso-butanol
	Dehydration of iso-butanol	Iso-butylene
	Intermediate steps	Bio-propylene
	Polymerization of bio-propylene	Bio-PP
PBS	Bacterial fermentation of glucose	Succinic acid
	Hydrogenation of succinic acid	1-4-butanediol
	Esterification between diacid (succinic acid) and diol (1–4, butanediol)	Butylene succinate
	Polycondensation at high temperature	PBS
PLA	ROP method	
	Fermentation of glucose or sucrose	Lactic acid
	Removal of condensed water, mesolactic acid, and low MW polymer	
	Recrystallization	High MW pure lactide
	Addition of coordination/insertion, cation, and anion (mechanisms initiators) to induce polymerization	Polylactic acid

**Table 2 polymers-15-02399-t002:** Comparison of reinforced bioplastics against conventional plastics and bioplastics.

Properties	Conventional Plastics	Bioplastics	Reinforced Bioplastics	References
Biodegradability	Non-biodegradable	Biodegradable and non-biodegradable (depends on the material)	Biodegradable and non-biodegradable (depends on the material)	[12,14,37,46,50]
Sustainability	No (Fossil-fuel based)	Yes (Renewable resources/ biomass)	Yes (Depends on the materials)	[33,34,51,52]
Mechanical strength	High (Can withstand heavy workload)	Low	Moderate—High (Depending on the reinforcement)	[27,52,53,54]
Water permeability	Low	High	Low—Moderate (Depending on the reinforcement)	[27,37,55,56,57]
Thermal stability	High	Low	Moderate—High (Depending on the reinforcement)	[27,52,53,58,59]
Brittleness	Low	High	Low	[52,53,54,59,60]

**Table 3 polymers-15-02399-t003:** The applications of bioplastics and reinforced bioplastics in various industries.

Industries	Types	Applications	References
Healthcare	Reinforced PLA	Gene delivery	[66,67,68]
		Tissue engineering	
		Implants	
		Shape memory	
		Controlled-release drug delivery	
Electrical and electronic	Bioplastics reinforced with CNT/cellulose nanofibre	Flexible photovoltaic cells (Solar cell)	[79,80]
		Sensors	
		Substrate in roll-to-roll fabrication techniques	
	Graphene reinforced PLA	Three-dimensional printing filament	[81]
Architecture and construction	Lignocellulosic fibre reinforced bioplastics	Window frames and doors	[89]
		Stabilisers for earthen construction materials	[90]
Agricultural	EFB reinforced bioplastics	Mulch film	[57,99]
		Plant nursery bags or pots	[100,101]
		Agricultural products packaging	[102]
Packaging	Bio-based ethylene glycol	PepsiCo packaging	[101,103]
	PHA, PLA and starch	Food packaging	[104]
	EFB or rice straw reinforced bioplastics	Fresh fruit packaging	[102]
	Fibre reinforced PHBV	Active food packaging	[107]

## Data Availability

Not applicable.

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
