# Peer review of "The Potential Applications of Reinforced Bioplastics in Various Industries: A Review"

_polymers, 2023, doi:10.3390/polym15102399_

Round 1

Reviewer 1 Report

The manuscript is a review of the literature on bioplastics.  Unfortunately, the entire document is written in a form of a qualitative description and NOT a quantitative study.  The manuscript does not list any numerical property of bioplastics (viz. strength, MPa, or modulus, GPa).

The manuscript was devoid of any quantitative analysis.

The conclusions were more of authors opinions as no numbers were presented in any of the sections.

There were NO quantitative tables or figures.

Author Response

Thank you so much for your valuable comment. We have amended our manuscript accordingly in page 7-8.

However, the conclusions were based on the information from the references, hence from our point of view, it is not appropriate to have quantitative analysis and to have number in this section.

Reviewer 2 Report

This article reviewed the potential application of reinforced bioplastics in various industries, which comprehensively summarized the latest publications and provided helpful guidance for such study. It can be accept after minor revision.

1.     The title should be changed as “ the potential application of reinforced bioplastics in …..”

2.     Bioplastics are really important in many fields. But I think It should be recycled firstly. In this respect, you should summarize the latest literature. For example, you can add some studies about recycled polymer of Prof. Qihua Wang from LICP, China.

3.     The traditional reinforced materials for polymer were CF, GF or aramid fibers. But for bioplastics, you should focus on the study about biofibre as reinforced materials. 

Author Response

Reviewer 2

Responses

The title should be changed as “ the potential application of reinforced bioplastics in …..”

Thank you for your comments and suggestions.

We have changed the title according to your suggestions as highlighted in line 2.

Bioplastics are really important in many fields. But I think it should be recycled firstly. In this respect, you should summarize the latest literature. For example, you can add some studies about recycled polymer of Prof. Qihua Wang from LICP, China.

Yes, agree with the reviewer.  We have read some studies about recycled polymer and added the summary in the manuscript including studies from Prof. Qihua Wang. (As highlighted in line 39 to line 48.

The traditional reinforced materials for polymer were CF, GF or aramid fibers. But for bioplastics, you should focus on the study about biofibre as reinforced materials. 

Biofibre indeed a strong reinforced materials and the focus of future study on reinforced bioplastics. In the manuscript, it more on studying the trend of different applications, therefore, it covered few types of reinforcing materials but not mainly focus on biofibre. However, to improve the quality of this manuscript, we have added some information about biofibre in page 7-8.

Reviewer 3 Report

The article discusses about bioplastics/reinforced bioplastics and their applications of bioplastics in various industries. However, the objective of the research is not clearly presented neither in the Abstract nor in the Introduction. In my opinion the article needs major improvements. A clearer and more creative presentation (especially) could add value to this article.

Some examples of improvements are presented below:

-          References must be cited in the text in according to the journal's requirements.  Exempli gratia Row 176 In the study of Fengge Gao (2004), correct: In the study of Gao [34]; Row 182 Shiou Xuan Tan et al. (2022), correct: Tan et al. [36] Please look for this aspect in the whole article.

-          Acronyms/Abbreviations should be defined only when it appears for the first time in the text. Then only the acronym is used in the text. Please look for this aspect in the whole article.

-          Please make the connection between Table 1 and the body of article.

-          I suggest you pay attention to the punctuation and the spaces between words. E.g. Row 120 2.2.1.. Natural; Row 172 [32,33] According to Satyanarayana; Row 218   2.3. . Comparison; Row 367 Bozó et al (2021), stated; Row 410 in these industries are, pipes. Please look for this aspect in the whole article.

-          Row 187 and abundant in nature c. - Is c necessary?

-          Please make the connection between Table 2 and the body of article.

-          Please rearrange Table 3 and create the link between it and the body of article.

-          The review article would be good to have at least one figure, of course which is related to the text.

-          It would be appropriate for the authors to present the research methodology. What databases did the authors study, how did the authors search for articles, on what basis did the authors select them. The number of articles accessed.

-          Although the conclusions are related to the text, I believe that they could be improved. For example, I would not end with this sentence "Thus, further research and study on reinforced bioplastics is necessary to widen the applications of reinforced bioplastics.”

-          For the the references appropriate, other sources could also be consulted (not necessarily to be quoted, but as inspiration for the figures): 

Stoica D., Alexe P., Ivan A.S., Moraru D.I., Ungureanu C.V., Stanciu S., Stoica M. 2022. Biopolymers: Global carbon footprint and climate change, In Nadda, A.K., Sharma, S., Bhat, R. (eds.) Biopolymers Recent Updates, Challenges and Opportunities. Springer Series on Polymer and Composite Materials. Springer, Cham. https://doi.org/10.1007/978-3-030-98392-5_3, pp 35–54.

Perhaps the authors have access to this book Biopolymers Recent Updates, Challenges and Opportunities. Springer Series on Polymer and Composite Materials. Springer, Cham. In which you will certainly find other aspects worthy of being visited.

-          Please include any additional comments on the tables and figures. First of all, the article does not include any figure. I think it would be appropriate to include at least one figure. I think it would be appropriate for the tables to be organized differently. To have a separate column for references, not to contain those arrows and numbers (Table 1). Only Bio-PE and Bio-PP are covered in the Table 1. Table 2 does not contain references for all the information presented (for example for sustainability).

Author Response

Reviewer 3

Responses

1.

a. References must be cited in the text in according to the journal's requirements.  Exempli gratia Row 176 In the study of Fengge Gao (2004), correct: In the study of Gao [34]; Row 182 Shiou Xuan Tan et al. (2022), correct: Tan et al. [36] Please look for this aspect in the whole article.

b. Acronyms/Abbreviations should be defined only when it appears for the first time in the text. Then only the acronym is used in the text. Please look for this aspect in the whole article.

c.  I suggest you pay attention to the punctuation and the spaces between words. E.g. Row 120 2.2.1.. Natural; Row 172 [32,33] According to Satyanarayana; Row 218   2.3. . Comparison; Row 367 Bozó et al (2021), stated; Row 410 in these industries are, pipes. Please look for this aspect in the whole article.

D. Row 187 and abundant in nature c. - Is c necessary?

Thank you for your comments and suggestions.

For all the references, acronyms, abbreviations, punctuation, spaces, and typo issues, I have checked for these aspects in the whole article, and I have made the amendment accordingly.

2.

Please include any additional comments on the tables and figures.

Necessary comments and elaboration on the tables as highlighted have added in line 569.

3.

Please make the connection between Table 1 and the body of article.

We have rearranged and added information on Table 1 to connect with the body of article, as suggested.

4.

Please rearrange Table 3 and create the link between it and the body of article.

Table 3 has been rearranged and linked to the body of article, as suggested.

5.

I think it would be appropriate for the tables to be organized differently. To have a separate column for references, not to contain those arrows and numbers (Table 1). Only Bio-PE and Bio-PP are covered in the Table 1. Table 2 does not contain references for all the information presented (for example for sustainability)

We have re-organised the table using separate column of the table as highlighted in table 1 and 2 (line 123 and 265 respectively. I have included other bioplastics covered in the manuscript into table 1. For the references issues in table 2, I have added the all the necessary references, as suggested.

6.

It would be appropriate for the authors to present the research methodology. What databases did the authors study, how did the authors search for articles, on what basis did the authors select them. The number of articles accessed.

Thanks for the suggestions again. As we have studied on other review articles, we could not find any of them emphasize on their methodology. Hence, we propose to remind as it is. However, we will appreciate it if the reviewer could enlighten us in this.

7.

Although the conclusions are related to the text, I believe that they could be improved. For example, I would not end with this sentence "Thus, further research and study on reinforced bioplastics is necessary to widen the applications of reinforced bioplastics.”

We have made some changes on the conclusion and added future prospects on the study of reinforced bioplastics, as suggested.

8.

First of all, the article does not include any figure. I think it would be appropriate to include at least one figure.

The review article would be good to have at least one figure, of course which is related to the text.

Thank you for your suggestion. As it is a review manuscript, it is a bit hard for us to have figure for comparison, which we strongly feel table is a better option.

9.

The article discusses about bioplastics/reinforced bioplastics and their applications of bioplastics in various industries. However, the objective of the research is not clearly presented neither in the Abstract nor in the Introduction.

We have stated the objectives in the Introduction, and added into the abstract as suggested  (highlighted in line 65-69 and line 24-26 respectively)

Round 2

Reviewer 3 Report

In my opinion, the article should be published.

However, relate to:

-          Please check the English language.

-          Row 2 – applications

-          Row 20 - Before 21st, st at superscript

-          Row 48 - Please add point at the end of the sentence.

-          Row 50 - A. Rudin and P. Choi [7] Capital letters A and P are not necessary. Please look for this aspect in the whole article.

-          Row 120 -  The point is not necessary before summarise.

-          Row 122 -  Valentina is not necessary. First name is not required. Please look for this aspect in the whole article.

-          Row 186 -  [36,37] Please add point. I suggest you pay attention to the punctuation in the hole article. Please look for this aspect in the whole article.

-          Please make the connection between Table 2 and the body of article.

-          I suggest you pay attention to the spaces between words in the hole articles.

-          Many references in the text are written in italics, some are underlined. Please correct.

Author Response

Reviewer 3

Responses

1.

- Please check the English language.

- Row 2 – applications

- Row 20 – Before 21st, st at superscript

- Row 48 – Please add point at the end of the sentence

- Row 50 – A. Rudin and P. Choi [7], capital letters A and P are not necessary. Please look for this aspect in the whole article

- Row 120 – The point is not necessary before summarise.

- Row 122 – Valentina is not necessary. First name is not required. Please look for this aspect in the whole article.

- Row 186 – [36,37] Please add point. I suggest you pay attention to the punctuation in the whole article. Please look for this aspect in the whole article.

- I suggest you pay attention to the spaces between words in the whole article.

- Many references in the text are written in italics, some are underlined. Please correct

Thank you so much for your valuable comment and pointing out the mistakes made in the manuscript. We have amended our manuscript accordingly.

Punctuation and spaces errors have been corrected accordingly throughout the whole article.

References / Citation issues have been amended accordingly.

The references in the text are all corrected.

2.

Please make the connection between Table 2 and the body of article.

Some descriptions on the table 2 are added, and the text is moved below the table.
